# Simulating the drought response of European tree species with the dynamic vegetation model LPJ-GUESS (v4.1, 97c552c5)

Benjamin F. Meyer[1, *], João P. Darela-Filho[1], Konstantin Gregor[1], Allan Buras[1], Qiao-Lin Gu[1], Andreas Krause[1], Daijun Liu[2], Phillip Papastefanou[3], Sijeh Asuk[4], Thorsten E. E. Grams[5], Christian S. Zang[6], and Anja Rammig[1]

[1]Technical University of Munich, Professorship of Land Surface-Atmosphere Interactions, TUM School of Life Sciences, Freising, Germany
[2]Department of Botany and Biodiversity Research, University of Vienna, Rennweg 14, 1030 Vienna, Austria
[3]Department Biogeochemical Signals, Max–Planck-Institute for Biogeochemistry, Hans-Knoll-Str., 10, Jena, 07745, Thuringia, Germany
[4]Department of Geography and Environment, School of Social Sciences and Humanities, Loughborough University, Loughborough, UK. LE11 3TU
[5]Technical University of Munich, Professorship of Land Surface-Atmosphere Interactions, Ecophysiology of Plants, TUM School of Life Sciences, Freising, Germany
[6]University of Applied Sciences Weihenstephan-Triesdorf, Professorship of Forests and Climate Change, Freising, Germany

**Correspondence:** Benjamin F. Meyer (ben.meyer@tum.de)

**Abstract.** Due to climate change severe drought events have become increasingly commonplace across Europe in recent decades with future projections indicating that this trend will likely continue, posing questions about the continued viability of European forests. Observations from the most recent pan-European droughts suggest that these types of "hotter droughts" may acutely alter the carbon balance of European forest ecosystems. Yet, substantial
uncertainty remains regarding the possible future impacts of severe drought on the European forest carbon sink. Dynamic vegetation models can help to shed light on such uncertainties, however, the inclusion of dedicated plant hydraulic architecture modules in these has only recently become more widespread. Such developments intended to improve model performance also tend to add substantial complexity, yet, the sensitivity of the models to newly introduced processes is often left undetermined. Here, we describe and evaluate the recently developed mechanistic
plant hydraulic architecture version of LPJ-GUESS and provide a parameterization for 12 common European forest tree species. We quantify the uncertainty introduced by the new processes using a variance-based global sensitivity analysis. Additionally, we evaluate the model against water and carbon fluxes from a network of eddy covariance flux sites across Europe. Our results indicate that the new model is able to capture drought-induced patterns of evapotranspiration along an isohydric gradient and manages to reproduce flux observations during drought better
than standard LPJ-GUESS. Further, the sensitivity analysis suggests that hydraulic process related to hydraulic failure and stomatal regulation play the largest roles in shaping model response to drought.

# 1 Introduction

For the past decades, the face of European forests has been increasingly marred by heatwaves and droughts – effects of anthropogenic climate change (Ciais et al., 2005; European Environment Agency, 2019; Bigler and Vitasse, 2021; Fink et al., 2004). Severe pan-European droughts in 2003, 2018, and 2022 in combination with record high temperatures (so-called "hotter droughts") caused record reductions in forest growth and productivity as a result of defoliation, higher susceptibility to biotic agents, and mortality (Buras et al., 2023; Ciais et al., 2005; Schuldt et al., 2020; van der Woude et al., 2023). Concerningly, the most recent carbon losses induced by the 2022 hotter-drought have turned central European forests from a carbon sink to a carbon source (van der Woude et al., 2023). With more frequent and intense droughts looming on the horizon, the future of the European forest carbon sink remains uncertain (Brodribb et al., 2020; Cook et al., 2020; Pan et al., 2024, 2011). While dynamic vegetation models (DVMs) are popular tools commonly used to shed light on such uncertainties and estimate possible future impacts on the vegetation carbon sink, many of the established models display strongly diverging simulations in regard to the effects of drought and heat (Tschumi et al., 2023). In an attempt to ensure that future vegetation changes and the associated feedbacks on the water and carbon cycles can be simulated confidently, the latest generation of dynamic vegetation models features increasingly detailed representations of plant hydraulic architecture (Xu et al., 2016; Yao et al., 2022; Kennedy et al., 2019; Xu et al., 2023; Eller et al., 2018, 2020; Christoffersen et al., 2016).

In the simplest terms, these representations of hydraulic architecture tend to consider two distinct drivers of drought induced stress: insufficient water availability in the soil and increased atmospheric demand for water (Papastefanou et al., 2020). The balance between supply and demand determines whether a tree will experience drought stress or not. The link between these two ends of the system is the hydraulic architecture of the tree which utilizes the xylem to transport water from the the roots through the stem and ultimately to the leaves where it is transpired through the stomata into the atmosphere (Lambers and Oliveira, 2019). Disruptions of this pipeline due to cavitation or stomatal closure trigger symptoms commonly associated with drought stress. As the ability of trees to transport water declines, other processes such as photosynthetic assimilation and growth cease (Lambers and Oliveira, 2019; Choat et al., 2012). Ultimately, critical dehydration – either directly or through predisposing affected trees to pathogens or insect attack – leads to tree death (Anderegg et al., 2012; Mcdowell et al., 2008; Hajek et al., 2022; Bigler et al., 2006).

Earlier DVMs generally included simple mechanisms to simulate drought stress, frequently opting for empirical approaches to reduce photosynthetic assimilation during periods of low water availability (Powell et al., 2013; Smith et al., 2001; Zhou et al., 2013). This strategy does not account for the mechanistic links between species-specific hydraulic traits, such as xylem vulnerability to cavitation, stomatal response to atmospheric drying, and xylem conductivity, which have been shown to play a key role in modulating the impact of drought conditions on forests in terms of both productivity and mortality (Hajek et al., 2022; Anderegg et al., 2016, 2015). To account for this behavior, current DVMs are increasingly including mechanistic, process-based representations of plant hydraulic architecture with functional diversity in regards to stomatal control, water-potential regulation, water-flow through

the soil-plant-atmosphere continuum, and hydraulic failure under drought conditions (Xu et al., 2016, 2023; Eller et al., 2018, 2020; Yao et al., 2022; Kennedy et al., 2019; Christoffersen et al., 2016; De Kauwe et al., 2020; Papastefanou et al., 2020, 2024).

While these improvements prove valuable in predicting the response of forests to present and future drought,
they add further complexity to already complex models by introducing new parameters and processes potentially contributing to increased uncertainty between projections from various models (Oberpriller et al., 2022; Zaehle et al., 2005). Identifying the causes of uncertainty can help guide future model development, highlight the need for more observations of key traits, and determine which model processes may be over- or underrepresented compared to reality (Zaehle et al., 2005; Dietze et al., 2018). In this context, global sensitivity analysis is commonly used to
detect the sensitivity of model outputs to model parameters (Saltelli, 2008). Due to the complexity of DVMs and the associated computational demand in performing a comprehensive global sensitivity analysis, such analyses are rare and not consistently applied each time new processes are implemented and new parameters are introduced (Oberpriller et al., 2022). Nevertheless, these analyses remain paramount for enhancing our understanding of the internal model processes and are invaluable in allowing solid interpretation of model results (Oberpriller et al., 2022;
Zaehle et al., 2005; Pappas et al., 2013).

Here, we describe and examine the recently developed mechanistic hydraulic architecture in LPJ-GUESS, termed LPJ-GUESS-HYD, intended to more accurately capture tree drought responses based on the theoretical framework of isohydricity (Papastefanou et al., 2024). The concept of isohydricity has been used to classify the response patterns of trees to drought (Tardieu et al., 2015; Jones and Sutherland, 1991) based in part on the sensitivity of leaf water
potential to changes in canopy conductance (Klein, 2014). LPJ-GUESS-HYD builds upon a previous version of LPJ-GUESS with mechanistic plant hydraulic architecture which, although it did not implement the impact of xylem cavitation and stomatal regulation related to isohydricity, nevertheless was able to reproduce patterns of potential natural vegetation (Hickler et al., 2006). LPJ-GUESS-HYD expands upon this earlier version by including a dynamic representation of species-specific water-potential regulation related to the concept of isohydricity (Papastefanou et al.,
2020) and explicitly coupling the model representation of evapotranspiration to canopy conductance governed by plant hydraulic processes (Papastefanou et al., 2024) in contrast to the standard version of LPJ-GUESS which only does so during periods of limited water availability.

To thoroughly evaluate the implemented processes related to drought induced stress and the sensitivity of the model to the model parameters governing these processes we conduct a variance-based global sensitivity analysis (Saltelli,
2008; Saltelli et al., 2010). To forego the limitations associated with the complexity of DVMs and the computational demand of running a sensitivity analysis, we focus on the newly introduced parameters governing the plant drought response. Accordingly, we compiled parameter ranges for 12 major European forest tree species from observations and analyzed their sensitivities by simulating a network of 34 eddy covariance flux sites throughout Europe (Warm Winter 2020 Team and ICOS Ecosystem Thematic Centre, 2022). Furthermore, we establish viable parameterizations

for our set of 12 species to compare simulated and observed evapotranspiration and gross primary productivity across the European forest sites.

We aim to answer the following questions:

1. Which of the seven newly introduced parameters related to hydraulic architecture introduce the most uncertainty to LPJ-GUESS-HYD?
2. Does the inclusion of hydraulic architecture reflect species-specific drought responses along an isohydricity gradient in the model, that is, under increasing drought will anisohydric species continue to transpire more than isohydric species?
3. Does LPJ-GUESS-HYD represent an improvement over LPJ-GUESS in depicting the drought response as represented by changes in GPP and evapotranspiration in European forest ecosystems when compared to observational data from eddy-covariance flux towers?

## 2 Methods

### 2.1 Description of standard version of LPJ-GUESS

LPJ-GUESS is a dynamic vegetation model simulating terrestrial ecosystem dynamics on a regional to global scale driven by atmospheric CO2, gridded meteorological inputs, nitrogen deposition, and soil physical properties (Smith et al., 2001, 2014). The model has been successfully applied and evaluated on global (e.g. Seiler et al., 2022) and regional scale (e.g. Hickler et al., 2012) for a wide range of applications in both managed (e.g. Lindeskog et al., 2021) and natural forest ecosystems (e.g. Ahlström et al., 2012). The following sections will provide an overview of LPJ-GUESS with particular focus on the model processes critical to the representation of drought effects on individual trees.

### 2.1.1 Representation of vegetation in LPJ-GUESS

Within each simulated gridcell or site, replicate patches serve as random samples of the entire landscape to account for disturbance- and stand-development-related differences between vegetation stands. Vegetation dynamics in each patch emerge from the competition of different age cohorts of plant functional types (PFTs) or species for space, light, water, and nutrients. Individuals within a cohort are identical in age and size. Typically, PFTs represent classes of tree species with similar attributes in regards to characteristics such as phenology, shade-tolerance, bioclimatic limits, etc., that are described by a common set of parameters. Here, we use the parameterization developed by Hickler et al. (2012) and expanded upon by Lindeskog et al. (2021) to simulate a subset of the most pertinent European tree species. Except for the newly introduced hydraulic parameters (Table 2), all species parameters are identical to those in Lindeskog et al. (2021).

LPJ-GUESS simulates photosynthesis and stomatal conductance based on the BIOME3 model (Sykes and Prentice, 1996) along with respiration, and phenology on a daily basis. At the end of each simulation year, accumulated net primary productivity (NPP) is allocated to leaves, roots, and sapwood following allometric constraints (Sitch et al., 2003). Population dynamics (establishment and mortality) and patch-destroying disturbances are simulated stochastically on a yearly time-step. Soil carbon and nitrogen cycles are simulated based on the CENTURY model

(Parton et al., 2010; Kirschbaum and Paul, 2002; Parton et al., 1993; Comins and McMurtrie, 1993).

### 2.1.2  Soil hydrology

Soil hydrology is represented as a "leaky bucket" model with percolation between layers based on Gerten et al. (2004) albeit with 15 soil layers (each 10 cm thick) instead of the original two (Zhou et al., 2024). The first five soil layers are considered "surface" layers and the remaining ten are referred to as "deep" layers. For each soil layer, $l$ (1 to 15),

the available water holding capacity ($awc_l$; mm) is determined by the volumetric water content at wilting point ($wp_l$; mm mm$^{-1}$), the (volumetric) field capacity ($fc_l$; mm mm$^{-1}$) and the soil layer thickness ($Dz_l$; mm) as:

$$\text{awc}_l = (fc_l - wp_l) * Dz_l \tag{1}$$

Field capacity and wilting point are determined by physical soil texture properties (e.g. clay-, sand-, and, silt-fraction, soil carbon content, bulk density, etc.) provided as input to the model and are the same for all layers. The dimensionless

ratio of $awc_l$ to the actual available liquid water in the soil ($aw_l$; mm) is defined as the water content ($wcont \in [0,1]$):

$$\text{wcont}_l = \frac{aw_l}{awc_l} \tag{2}$$

which indicates the amount of water available to plants in any given soil layer. Water input to soil comes from rainfall and snowmelt which are initially distributed among the five surface layers and subsequently percolate to the deeper layers. Water leaves the soil via evapotranspiration – where evaporation occurs from the fraction of soil not covered

by vegetation and transpiration is dependent on vegetation characteristics – and runoff.

### 2.1.3  Water availability dynamics

In the standard version of LPJ-GUESS only a few processes are limited by water availability, but the plant hydraulic architecture is not explicitly modeled. Nevertheless, certain processes are affected by limited water availability reflecting plant responses to drought. Initially, low water availability – drought – constrains the establishment of

new plant individuals. Each species is assigned a drought tolerance level from 0 (extremely drought tolerant) to 1 (extremely drought intolerant) based on the water content as a fraction of the available water holding capacity

required for that species to establish. This tolerance level is compared to the growing season average water content integrated over the upper five soil layers:

$$establish = \begin{cases} \text{false,} & \text{drought\_tolerance} > \text{wcont} \\ \text{true,} & \text{drought\_tolerance} \leq \text{wcont} \end{cases} \tag{3}$$

Additionally, drought can limit photosynthetic assimilation by downregulating canopy conductance ($g_c$; mm s$^{-1}$) and restricting the ratio ($\chi_{CO_2}$) of inter-cellular $CO_2$ ($c_i$; ppm) to ambient $CO_2$ ($c_a$; ppm) (Haxeltine and Prentice, 1996). Photosynthesis is modeled based on the Collatz simplification of the Farquhar model (Collatz et al. 1991) described in detail in Haxeltine and Prentice (1996) and Sitch et al., (2003). When water supply is ample, the optimal canopy conductance for photosynthesis is calculated as:

$$g_p = g_{min} + \frac{1.6 * A_{dt}}{c_a * (1 - \lambda_{max})} \tag{4}$$

where $g_{min}$ is the species-specific minimum canopy conductance (a parameter), $A_{dt}$ is the daytime net assimilation, and $\lambda_{max}$ is a species-specific parameter. Conversely, when water supply is limited, photosynthesis is calculated using *actual* rather than *maximum potential* canopy conductance, $g_p$, and $\chi_{CO_2}$ where the *actual* canopy conductance, $g_c$, is calculated as:

$$g_c = -g_{min} * \ln\left[\frac{1 - E_{su}}{E_q * \alpha_m}\right] \tag{5}$$

where $E_q$ is the equilibrium transpiration (mm s$^{-1}$), $E_{su}$ is the water supply (mm s$^{-1}$), and $\alpha_m$ is an empirical parameter (Haxeltine and Prentice, 1996). This calculation is triggered under water-stressed conditions, i. e. when the supply of water from the soil ($E_{su}$; mm s$^{-1}$) determined by the species-specific maximum transpiration rate ($E_{max}$, a species-specific parameter; mm s$^{-1}$) and the the soil moisture availability in the rooting zone ($W_r$, mm s$^{-1}$), that 160 is, the fraction of soil water content accessible to an individual based on the parameterized species-specific root distribution across all soil layers (Haxeltine and Prentice, 1996):

$$E_{su} = E_{max} * W_r \tag{6}$$

is not sufficient to satisfy demand indicated by $E_{de}$:

$$E_{de} = \frac{E_q * \alpha_m * g_c}{g_c + g_m} \qquad (7)$$

Consequently, $g_c$ is reduced to ensure that plant transpiration ($E$, mm s$^{-1}$) matches the supply ($E_{su}$) such that:

$$E = \min\{E_{su}, E_{de}\} \qquad (8)$$

## 2.2 Description of hydraulic architecture as implemented in LPJ-GUESS-HYD

LPJ-GUESS-HYD provides a more in-depth implementation of plant physiological processes related to water availability (Papastefanou et al., 2024). Strategies for water-potential regulation along the isohydric spectrum determine how species react to changes in soil water availability (Papastefanou et al., 2020). The resulting water potential gradient governs the flow of water through the plant and, based on Darcy's law (Whitehead, 1998), determines the supply of water available for transpiration (Hickler et al., 2006). Atmospheric demand for water is driven by vapor pressure deficit (VPD) and, together with the supply of water, ultimately governs canopy conductance for photosynthetic assimilation. Lastly, to model the impact of drought on tree mortality, LPJ-GUESS-HYD includes an empirical representation of hydraulic failure mortality based on xylem cavitation. These new processes seamlessly integrate into the existing structure of LPJ-GUESS and primarily replace empirical relationships between soil hydrology and photosynthetic assimilation (Fig. 1).

### 2.2.1 Water-potential regulation

LPJ-GUESS-HYD incorporates the dynamic model for water-potential regulation introduced by Papastefanou et al. (2020). This model operates on the principle that water transport from the roots through the stem to the leaves and into the atmosphere is dictated by a dynamically changing forcing pressure ($\Delta\psi(t)$, MPa):

$$\Delta\psi(t) = \psi_l(t) - \psi_s(t) - \rho g h \qquad (9)$$

where, $\psi_s(t)$ (MPa) and $\psi_l(t)$ (MPa) are the respective soil and leaf water potential at time $t$. The gravitational pull is defined by $\rho * g * h$, with $\rho$ (kg m$^{-3}$) referring to the density of liquid water, $g$ (m s$^{-2}$) the gravitational acceleration, and $h$ (m) the canopy height. In situations with ample soil water supply $\Delta\psi$ is denoted as $\Delta\psi_{max}$, a parameter describing the average forcing potential under well-watered conditions.

Soil water potential is initially calculated as a function of soil water content according to (Saxton et al., 1986) for each soil layer $ly$:

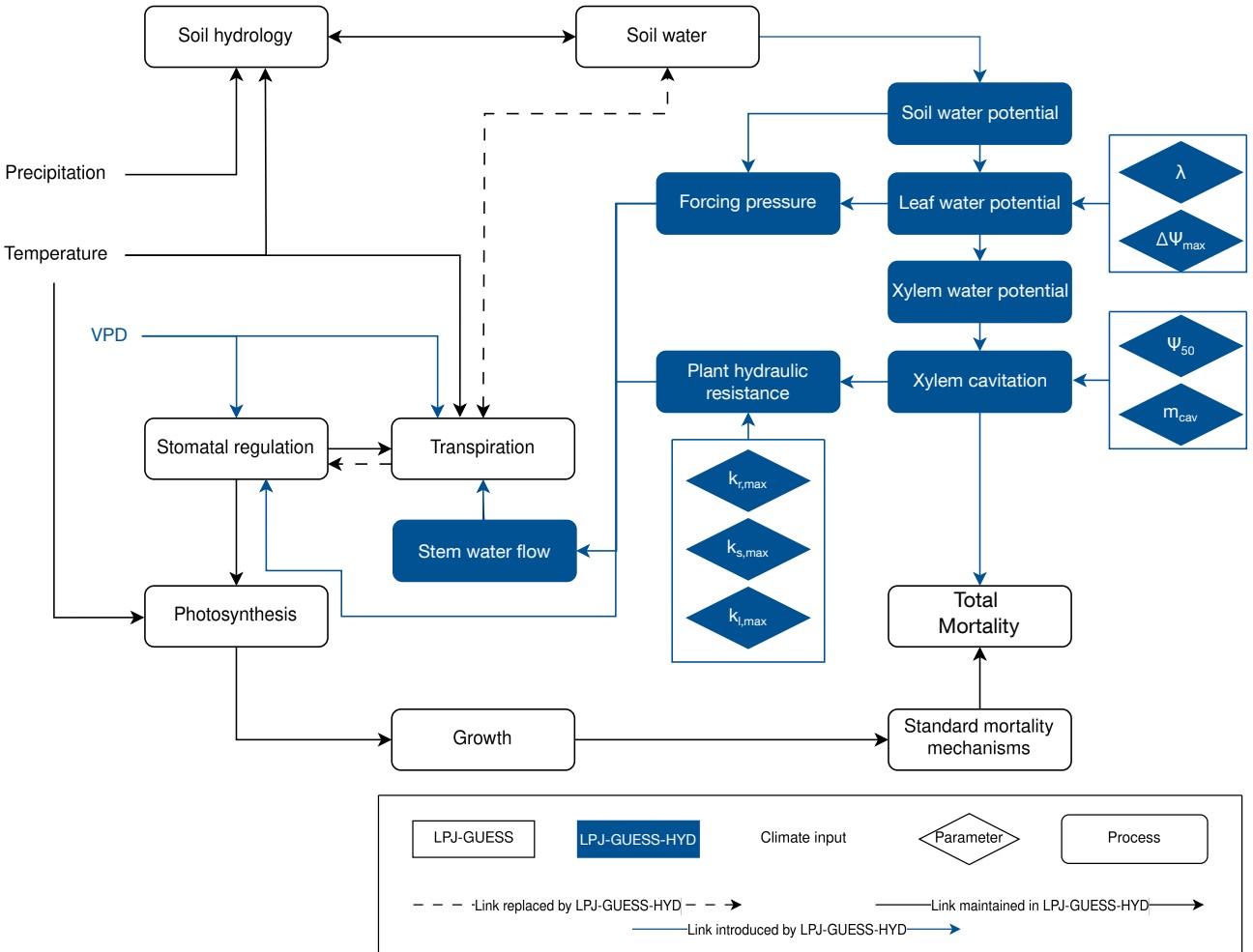

**Figure 1.** Flow chart displaying the model structure of LPJ-GUESS-HYD including links to standard LPJ-GUESS processes. Objects in blue are introduced by LPJ-GUESS-HYD while objects outlined in black are part of the standard LPJ-GUESS structure. Lines between boxes identify links between individual process, drivers, and parameters. Arrows indicate directionality. Dotted lines highlight links between processes in LPJ-GUESS which are replaced by an alternative structure in LPJ-GUESS-HYD. The light blue diamonds indicate the hydraulic parameters introduced by LPJ-GUESS-HYD and defined in Table 1.

$$\psi_{s_{ly}} = A * \text{wcont}_{tot,ly}^B \tag{10}$$

where $A$ and $B$ are functions of soil physical properties such as the clay and sand content (Eqs. A1 and A2; and see Saxton et al. (1986); ) and $\text{wcont}_{tot}$ (mm) is the sum of plant available soil water and the soil water content at wilting point.

Subsequently, $\psi_{s_{ly}}$ is weighted by the fraction of roots in that layer ($r_{f_{ly}}$) to give the integrated soil water potential $\psi_s$:

$$\psi_s = \sum_{l=0}^{N_l-1} \psi_{s_l} * r_{f_l} \tag{11}$$

where $N_l$ is the number of soil layers.

The model assumes that the change in $\psi_l$ over time depends on the difference between $\psi_l$ and $\psi_s$ such that:

$$\frac{d\psi_L}{dt} = \alpha((1-\lambda) * \psi_s(t) - \psi_l(t)) - \Delta\psi_{max} \tag{12}$$

where, $\lambda \in [0,1]$ (Table 1) is a component of the isohydricity of water potential regulation, with higher $\lambda$ contributing to more isohydric behavior and $\alpha$ (d$^{-1}$) is a rate parameter controlling how quickly $\psi_L$ adjusts to changes in $\psi_s$. As LPJ-GUESS-HYD runs on a daily timestep, $\alpha$ is set to 1 (Papastefanou et al., 2024). To account for summergreen phenology we expand upon Equation 12 to include the daily phenological status:

$$\hat{\psi}_l = \min\{\psi_l * \text{phen}, \psi_s\} \tag{13}$$

where *phen* is the leaf phenological status as a fraction of full leaf cover from 0 (no leaves) to 1 (full leaf cover). Subsequently, $\hat{\psi}_l$ equals $\psi_s$ during winter dormancy. For evergreens, *phen* is always 1 and thus $\hat{\psi}_l = \psi_l$.

Next, we assume that the stem-xylem water potential ($\psi_x$; MPa) is a function of $\hat{\psi}_l$ following Fisher et al. (2006):

$$\psi_x = b * (\hat{\psi}_l - \psi_s) + \psi_s + \rho g h \tag{14}$$

where $b$ represents the ratio of resistance belowground ($R_{bg}$; m$^2$ MPa s kg$^{-1}$) to total plant resistance ($R_p$; m$^2$ MPa s kg$^{-1}$):

$\quad b = \dfrac{R_{bg}}{R_p}$ (15)

### 2.2.2 Water supply in LPJ-GUESS-HYD

LPJ-GUESS-HYD simulates the effect of hydraulic architecture on water transport through the plant by using alternative formulations of $E_{su}$ and $E_{de}$ (from Equation 8).

The calculation of $E_{su}$ is adopted from Hickler et al. (2006):

$\quad E_{su} = \dfrac{-\Delta\psi}{R_r + R_s + R_l}$ (16)

where $R_r$, $R_s$, and $R_l$ are the hydraulic resistances of roots, stem, and leaves in m$^2$ MPa s kg$^{-1}$, respectively and are defined as:

$$R_r = \dfrac{1}{k_{r,max} * (1 - \text{plc}_\text{r}) * \frac{a_r}{h_{soil}} * \eta_s} \qquad (17)$$

$$R_s = \dfrac{1}{k_{s,max} * (1 - \text{plc}_\text{s}) * \frac{a_s}{h} * \eta_a} \qquad (18)$$

and

$$R_l = \dfrac{1}{k_{l,max} * (1 - \text{plc}_\text{l}) * a_l * M_{H_2O}} \qquad (19)$$

where $k_{r,max}$ (kg m$^{-1}$ s$^{-1}$ MPa$^{-1}$), $k_{s,max}$ (kg m$^{-1}$ s$^{-1}$ MPa$^{-1}$), and $k_{l,max}$ (mmol m$^{-2}$ s$^{-1}$ MPa$^{-1}$) are species-specific parameters describing the maximum potential conductance of each compartment (Table 1), $plc_r$, $plc_s$, and $plc_l$ are the fraction of cavitated vessels of each compartment, $a_s$, $a_r$ and $a_l$ are the cross-sectional area of sapwood, roots and

leaves in m$^2$ m$^{-2}$, respectively, $\eta_a$ and $\eta_n$ are the viscosity of water in the stem and soil, respectively, $h$ (m) is the tree height, $h_{soil}$ is the depth of the simulated soil column, $fpc$ is the individual's foliar projective cover and $M_{H_2O}$ is the molar mass of water (mol kg $^{-1}$).

The sum of resistances, denoted as $R_p$, represents the total plant hydraulic resistance:

$$R_p = R_r + R_s + R_l \qquad (20)$$

### 2.2.3 Water demand in LPJ-GUESS-HYD

The updated representation of $E_{de}$ is based on the instructive form of the Penman-Monteith equation described by Köstner et al. (1992) as:

$$E_{de} = \Omega * E_q + (1 - \Omega) * E_{imp} \tag{21}$$

$E_{imp}$ is the transpiration rate imposed by the effects of VPD, defined as:

$$E_{imp} = \frac{g_c * VPD}{\rho * G_v * T_{air}} \tag{22}$$

where $G_v$ (m$^3$ kPa kg$^{-1}$ K$^{-1}$) is the gas constant for water vapor and $T_{air}$ (K) is the ambient air temperature. The term $\Omega$ is the degree of coupling between the canopy and the atmosphere (i.e. VPD) representing the leaf/canopy boundary layer, defined as:

$$\Omega = \frac{1 + \varepsilon}{1 + \varepsilon + \frac{g_a}{g_c}} \tag{23}$$

where $\varepsilon$ is the change of latent heat relative to the change in sensible heat in air at 10 °C and $g_a$ (m s$^{-1}$) is the aerodynamic conductance. Consistent with the new formulations of $E_{de}$ and $E_{su}$, the calculation of $g_c$ is also updated. The assumption of the supply-demand principle underlying the original calculation of $g_c$ remains but the new definition reflects the dependence of plant water transport on VPD and hydraulic architecture. This is obtained by equating $E_{su}$ (Eq. 16) and $E_{imp}$ (Eq. 22) and solving for $g_c$ resulting in:

$$g_c = \frac{\lambda_{lvh} * \frac{\gamma}{cp_{air}} * \frac{\rho}{\rho_{air}} * \frac{\Delta \psi}{R_p}}{VPD} \tag{24}$$

where $\lambda_{lvh}$ (kJ kg$^{-1}$) is the latent heat of vaporization of water, $\gamma$ (kPa K$^{-1}$) is the psychrometric constant, $cp_{air}$ (kJ kg$^{-1}$ K$^{-1}$) is the specific heat of air, and $\rho_{air}$ (kg m$^{-3}$) is the density of air.

Subsequently, when the canopy conductance constrained by plant hydraulic processes (Eq. 24) is less than the non-stressed canopy conductance (Eq. 4) trees experience water limitation:

$$\text{water\_limitation} = \begin{cases} \text{true,} & g_c < g_p \\ \text{false,} & g_c \geq g_p \end{cases} \tag{25}$$

In this case, $g_c$ (Eq. 24) rather than $g_p$ (Eq. 4) is used in the photosynthesis calculation.

Through this representation of water supply and demand the integrity of the plant's water transport system can directly affect the canopy conductance and, subsequently, carbon assimilation through photosynthesis.

### 2.2.4  Cavitation and mortality

The transport of water from the soil through the plant and into the atmosphere described by (Eqs. 9 - 24) is susceptible to partial or total collapse when soil water availability (Eq. 2) does not suffice to satisfy the transpiration demand (Eq. 21). During periods of water limitation when evapotranspiration outweighs water availability, soil water potential declines (Eq. 10). Modulated by the species-specific hydraulic strategy ($\lambda$, $\Delta\psi_{max}$), leaf and xylem water potential react, as well (Equation 12). As $\psi_s$, $\psi_x$, and $\psi_l$ decrease, conductance through the tree (Eqs. 17 - 19)
is attenuated through higher resistance, stemming from the onset of cavitation. Cavitation is represented as the percentage loss of conductance ($plc$) in dependence on $\psi_x$, modeled as a sigmoidal curve (cf. Tyree et al., 1994; Tyree and Sperry, 1989; Sperry et al., 1998; Pammenter and Van Der Willigen, 1998):

$$\text{plc} = \frac{1}{\frac{\psi_x}{\psi_{50}}^{m_{cav}} + 1} \tag{26}$$

where $\psi_{50}$ (MPa) and $m_{cav}$ (MPa) are species-specific parameters indicating the xylem water potential at which 50 %
of conductance is lost and the slope of the vulnerability curve, respectively (Table 1). The slope parameter, $m_{cav}$, is calculated as:

$$m_{cav} = \frac{2}{log10(\frac{\psi_{50}}{\psi_{88}})} \tag{27}$$

where $\psi_{88}$ (MPa) is the water potential at which 88 % of conductance is lost. To curb drought-induced cavitation during winter when processes related to hydraulic failure are assumed to play only a minor role, cavitation is
only allowed to occur when $g_c$ is greater than $g_{min}$, the component of canopy conductance not associated with photosynthesis. With rising $plc$ the ability of plants to transport water is increasingly inhibited and eventually reaches a point of no return at which the inability to move water becomes lethal (Hammond et al., 2019; Wagner et al., 2023). The probability of fatal hydraulic failure ($p_{mort}$) is modeled as a Weibull function following the results from Hammond et al. (2019):

$$p_{mort} = 1 - e^{-\left(\frac{\text{plc}}{k_w}\right)^{\lambda_w}}$$

| Parameter | Unit | Min | Max | Data reference | Definition |
|---|---|---|---|---|---|
| $\psi_{50}$ | MPa | -14.20 | -0.11 | Choat et al. (2012) | Xylem pressure inducing 50% loss of conductance |
| $m_{cav}$ | MPa | -69.25 | -0.84 | Choat et al. (2012) | Slope of vulnerability curve between $\psi_{50}$ and $\psi_{88}$ |
| $k_{r,max}$ | kg m$^{-1}$ s$^{-1}$ MPa$^{-1}$ | 0.07 | 32.76 | Choat et al. (2012) | Maximum specific root conductivity |
| $k_{s,max}$ | kg m$^{-1}$ s$^{-1}$ MPa$^{-1}$ | 0.10 | 49.00 | Choat et al. (2012) | Maximum specific stem conductivity |
| $k_{l,max}$ | mmol m$^{-2}$ s$^{-1}$ MPa | 0.94 | 43.10 | Multiple sources[1] | Maximum specific leaf conductivity |
| $\lambda$ | - | -0.30 | 1.00 | Papastefanou et al. (2020) | Isohydricity scalar |
| $\Delta\psi_{max}$ | MPa | 0.26 | 4.46 | Papastefanou et al. (2020) | Forcing pressure under well watered conditions |

**Table 1.** Definitions of the 7 new hydraulic parameters introduced in LPJ-GUESS-HYD and the parameter ranges used in the sensitivity analysis. These ranges extend beyond the observed values for the 12 species used in this study in order to explore the model's reaction to as wide of a parameter range as plausible. The data reference column indicates the source of the compiled ranges for each parameter. [1]Flexas et al. (2013); Méndez-Alonzo et al. (2019); Johnson et al. (2009); Scoffoni et al. (2011); Johnson et al. (2016); Nolf et al. (2015); Blackman et al. (2010)

where $k_w$ is a shape parameter and $\lambda_w$ is a scale parameter. As *plc* approaches 100 %, i.e. total hydraulic failure, the probability of mortality tends toward 1.

## 2.3 Global Sensitivity Analysis

The new processes integral to LPJ-GUESS-HYD introduce seven new input parameters. To ascertain how these additions contribute to uncertainty in the model output, we perform a global sensitivity analysis on the new parameters. LPJ-GUESS simulates a large number of outputs suited for sensitivity analysis. Similarly to Oberpriller et al. (2022), we examine carbon- and water- related outputs (evapotranspiration, canopy conductance, NPP, and biomass) due to the importance of forests in the carbon cycle both in governing fluxes and contributing to the carbon sink, and in the water cycle (Bonan, 2008; Pan et al., 2011; Pugh et al., 2019). We place a strong focus on water-related outputs due to the role of water use in modulating forest productivity, particularly under drought conditions (Lambers and Oliveira, 2019; Sulman et al., 2016). Sensitivities were calculated by sampling parameter sets from the multivariate parameter space using Latin Hypercube Sampling (LHS) (Helton and Davis, 2003; Mckay et al.). LHS is a sampling technique which stratifies a parameter into equal, non-repeating intervals across its entire range. By randomly sampling with each interval, LHS reduces bias and efficiently ensures full coverage of the parameter space. Compared to other sampling techniques (e.g. Quasi-Random Numbers) LHS requires fewer samples to depict the "true" mean of the parameter range. Consequently, fewer simulations must be run, substantially reducing the computational effort required when working with complex models such as LPJ-GUESS-HYD (Saltelli, 2008). For each of the seven parameters we estimated the potential parameter range based on previous studies using all values for species classified as trees in the corresponding data sources (Table 1).

Subsequently, we created 6000 parameter sets via LHS covering the entire multivariate parameter space. The parameter sets were recycled for each of the 12 species and 34 sites.

We chose Sobol' indices to analyze the influence of parameter variations on the model output. This variance-based method can capture non-linear processes and is particularly suitable for non-additive models, i.e., models with interaction effects between the individual parameters such as the one (i.e. LPJ-GUESS-HYD) investigated here

Saltelli (2008). To calculate the sensitivity indices, LPJ-GUESS outputs needed to be condensed to a singular value per simulation (i.e. per parameter set). Flux variables (gross primary productivity (GPP), evapotranspiration, canopy conductance) were averaged over all years in the simulation period while the last year of the simulation was used for biomass. We calculated three sensitivity indices for each combination of output variable, species, and site. First and second order estimates were calculated using the estimator method introduced by (Saltelli et al., 2010). Total order

indices were computed following the method by Jansen (1999). First order indices measure the contribution of a single parameter to the variance in the model output excluding any interactions with other parameters. Similarly, second order indices measure the contribution of the interaction between two parameters to the variation in model output. Lastly, total order indices measure the contribution of a single parameter, including all its interactions with other parameters, to variation in the model output (Saltelli, 2008). In practical terms, these interactions refer to instances

where separate parameters jointly affect a given model process or a given model output. For example, leaf water potential regulation in LPJ-GUESS-HYD (Eq. 12) is driven in part by both $\lambda$ and $\Delta\psi_{max}$. In this case, the first order index for each parameter quantifies that parameter's individual contribution to Equation 12. The second-order index then quantifies the joint effect of the two parameters on Equation 11. This concept also extends beyond single, self-contained processes. That is, since, for example, both the water potential gradient between leaf and soil (governed

by $\lambda$ and $\Delta\psi_{max}$, Eq. 9, Eq. 11, Eq. 12) and the total plant resistance (governed by $k_{r,max}$, $k_{s,max}$, and $k_{l,max}$, Eq. 17)-19 affect canopy conductance the joint effect of any combination of these five parameters on canopy conductance can be quantified using either the second-order or total-order indices. The sensitivity indices range between 0 (least influential) and 1 (most influential) and depict the proportion of variance in the model output attributed to variations in a a given parameter or interaction of parameters. By sampling the parameters independently of one another, i.e.

allowing each parameter to vary independent of any other parameter in the same parameter set, we avoid collinearity biasing the sensitivity indices. To establish significance we calculated sensitivity indices for a dummy parameter (i.e. a parameter that has relationship to the model). First and second order indices for our analyzed parameters were considered significant only if their value was higher than the indices for the dummy parameter. We used the `sensobol` R package to sample the 6000 parameter sets and compute the sensitivity indices (Puy et al., 2022).

**2.4   Simulation Protocol and model evaluation**

To test the functionality of LPJ-GUESS-HYD across a wide range of species we selected 12 common forest tree species from boreal, temperate, and Mediterranean ecosystems and extracted the relevant parameters from available

| Species | $\Psi_{50}$ | $m_{cav}$ | $\Delta\psi\_ww$ | $\lambda$ | $k_{r,max}$ | $k_{s,max}$ | $k_{l,max}$ | Sites |
|---|---|---|---|---|---|---|---|---|
| *Abies alba* | -3.65 | -10.7 | 0.4 | 0.4 | 0.86 | 0.38 | 33.1 | 4 |
| *Betula pendula* | -2.23 | -10.96 | 1.15 | 0.4 | 1.12 | 1.86 | 19.54 | 1 |
| *Carpinus betulus* | -3.75 | -13.75 | 0.89 | 0.07 | 1.8 | 2.7 | 19.54 | 2 |
| *Fagus sylvatica* | -2.6 | -9 | 1.47 | -0.08 | 1.22 | 1.83 | 34.2 | 8 |
| *Fraxinus excelsior* | -2.8 | -7.95 | 0.78 | 0.45 | 0.47 | 0.7 | 8.88 | 1 |
| *Picea abies* | -3.7 | -12 | 1.15 | 0.4 | 0.29 | 0.43 | 33.1 | 17 |
| *Pinus halapensis* | -3.57 | -10.95 | 0.47 | 0.44 | 0.35 | 0.52 | 12.5 | 1 |
| *Pinus sylvestris* | -3.14 | -6.96 | 0.63 | 0.8 | 0.3 | 0.45 | 12.5 | 9 |
| *Populus tremula* | -1.65 | -6.67 | 0.86 | 0.53 | 0.61 | 0.92 | 25.39 | 1 |
| *Quercus ilex* | -3.27 | -4.77 | 1.14 | 0.16 | 1.3 | 1.95 | 7.95 | 5 |
| *Quercus pubescens* | -2.475 | -3.88 | 1.71 | 0.18 | 1.05 | 1.65 | 7.3 | 1 |
| *Quercus robur* | -2.8 | -9.45 | 1.6 | 0.075 | 2.05 | 2.34 | 9.9 | 9 |

**Table 2.** Best estimate species values for the 7 hydraulic parameters introduced in LPJ-GUESS-HYD used in the comparison of LPJ-GUESS-HYD with the eddy covariance flux variables. For each species, the used value is the mean of all values present for that species extracted from the relevant database (see Table 1). Where no observation for a given species was available, the genus mean was used instead.

plant trait databases (Table 2). Using data from plant trait databases for parameterizing models can carry potential pitfalls due to various methods used in the original analyses contributing the data (Cochard et al., 2013). To account

for this we ran an additional simulation using the same parameters as displayed in Table 2 but with $\psi_{50}$ values from Martin-StPaul et al. (2017) where such artifacts have been removed.

We chose sites to simulate from the ICOS Warm Winter 2020 ecosystem eddy covariance flux due to the availability of observational data for evaluation of the model at those sites (Warm Winter 2020 Team and ICOS Ecosystem Thematic Centre, 2022). We selected sites at which at least one of the 12 target species was present. This yielded

34 individual sites, each of which included a varying number of species, yielding a total of 55 unique species-site combinations. To avoid confounding effects brought on by competition between species each species at each site was simulated separately. For the sensitivity analysis we repeated the simulation of each species-site combination for all 6000 parameter sets. For evaluation of the model against the eddy-covariance flux data we used a set of best estimate parameters compiled from published literature for each species (Table 2). The forcing data and general simulation

procedure was the same for both sets of simulations.

The simulation period was from 1989 to 2020. To ensure a near-equilibrium state of the simulated ecosystem at the start of the simulation period we spun up the model for 1000 years by recycling the first 30 years of the climate inputs following standard procedure for LPJ-GUESS.

We forced both LPJ-GUESS and LPJ-GUESS-HYD with ERA-Interim daily mean surface temperature, precipitation
sum, shortwave radiation, average windspeed, pressure, and specific humidity which were downscaled to the specific
site coordinates and provided with the eddy-covariance flux data (Warm Winter 2020 Team and ICOS Ecosystem
Thematic Centre, 2022; Pastorello et al., 2020). Atmospheric $CO_2$ concentration were taken from NOAA (Lan et al.,
2023) and nitrogen deposition data were taken from Lamarque et al. (2011). Physical soil properties (e.g. clay-, sand-,
and, silt-fraction, soil carbon content, bulk density, etc.) were taken from the Harmonized World Soil Database v2.0
and aggregated by mode to match the 0.5° by 0.5° spatial resolution of the climate inputs (Har, 2023).

From the ICOS Warm Winter 2020 dataset we extracted the daily GPP averaged from half-hourly data and partitioned
via the night time partitioning method and daily evapotranspiration derived from observed latent heat flux (Allen
et al., 1998) to evaluate simulated GPP and evapotranspiration against (Pastorello et al., 2020).

## 3 Results

### 3.1 Sensitivity analysis

Of the seven parameters introduced in LPJ-GUESS-HYD, only two ($\psi_{50}$ and $\Delta\psi_{max}$) consistently contributed to
variance across various model outputs (Figure 2). Carbon mass in vegetation was most sensitive to variations in
$\psi_{50}$. Across all sites and species, the median contribution of $\psi_{50}$ to variation in carbon mass in vegetation, including
all interactions with other parameters, was 93.2% (Fig. 2a). Excluding any interactions with other parameters,
75% of the variance in carbon mass in vegetation was attributable solely to $\psi_{50}$ (Fig. 3a). Considering all possible
interactions, $\Delta\psi_{max}$ and $k_{l,max}$ were the second- (37.7%) and third-most (9%) influential parameters for carbon
mass in vegetation, respectively. However, no substantial first-order influence of either $\Delta\psi_{max}$ or $k_{l,max}$ was found
(Figure 3a). Generally, the analysis revealed similar patterns of total order sensitivity for GPP and evapotranspiration.
In all cases, $\psi_{50}$ contributed the most to variability in the output. Larger differences only manifested themselves in
the sensitivity of canopy conductance. While canopy conductance only showed significant first-order sensitivity to
$\psi_{50}$, it displayed a number of significant second-order sensitivities (Figure 3c). Additionally, all sensitivity indices
(total, first, and second) displayed a larger spread across species and sites for canopy conductance than for any of
the other variables (Figure 2d; Figure 3d). Importantly, while the sensitivity indices for $psi_{50}$ by far outweighed
those of the other parameters for GPP, evapotranspiration and vegetation carbon, the relative sensitivity of canopy
conductance to $psi_{50}$ compared to the other parameters was more balanced.

Although the total-order indices indicated that $m_{cav}$ contributed only marginally to output variance, the first-order
indices revealed that $m_{cav}$ on its own did, in fact, lead to significant albeit low variance in all model outputs (Figure 3).
For all considered output variables, second-order interactions consistently included $\psi_{50}$ and $\Delta\psi_{max}$ (Figure 3), while
only two other parameters, $k_{l,max}$ and $k_{r,max}$, occasionally featured in the second-order indices (Figure 3).

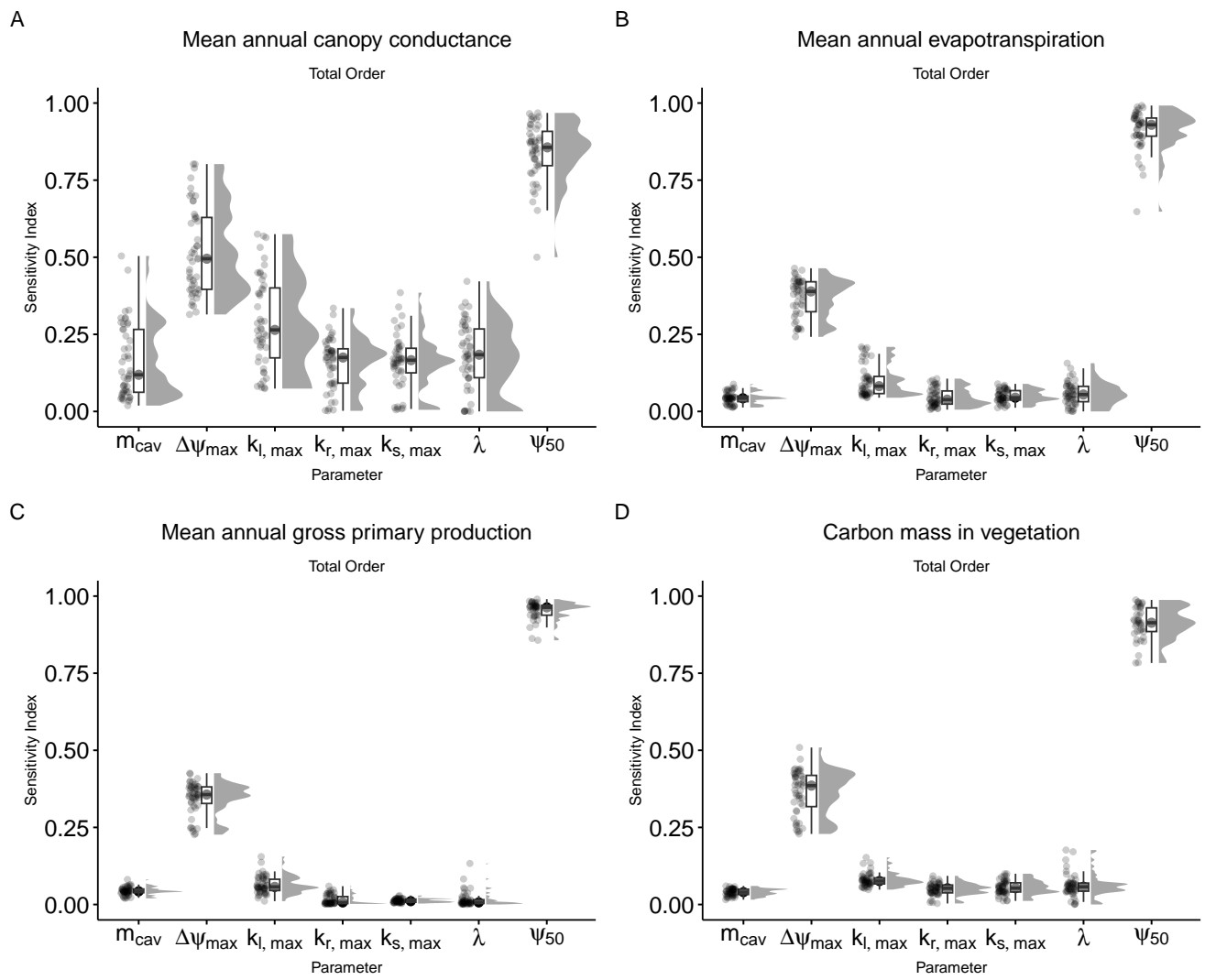

**Figure 2.** Total order sensitivity indices for the seven parameters introduced in LPJ-GUESS-HYD. Total-order indices indicate the sensitivity of model output to variation of a given parameter including any and all interactions with other parameters. Each point represents the sensitivity index for a single species-site combination. The boxplots indicate the median and interquartile range of the sensitivity indices across species-site combinations. Each panel shows the sensitivity indices for a single model output (A) mean annual canopy conductance , (B) mean annual evapotranspiration, (C) mean annual gross primary productivity, (D) carbon mass in vegetation.

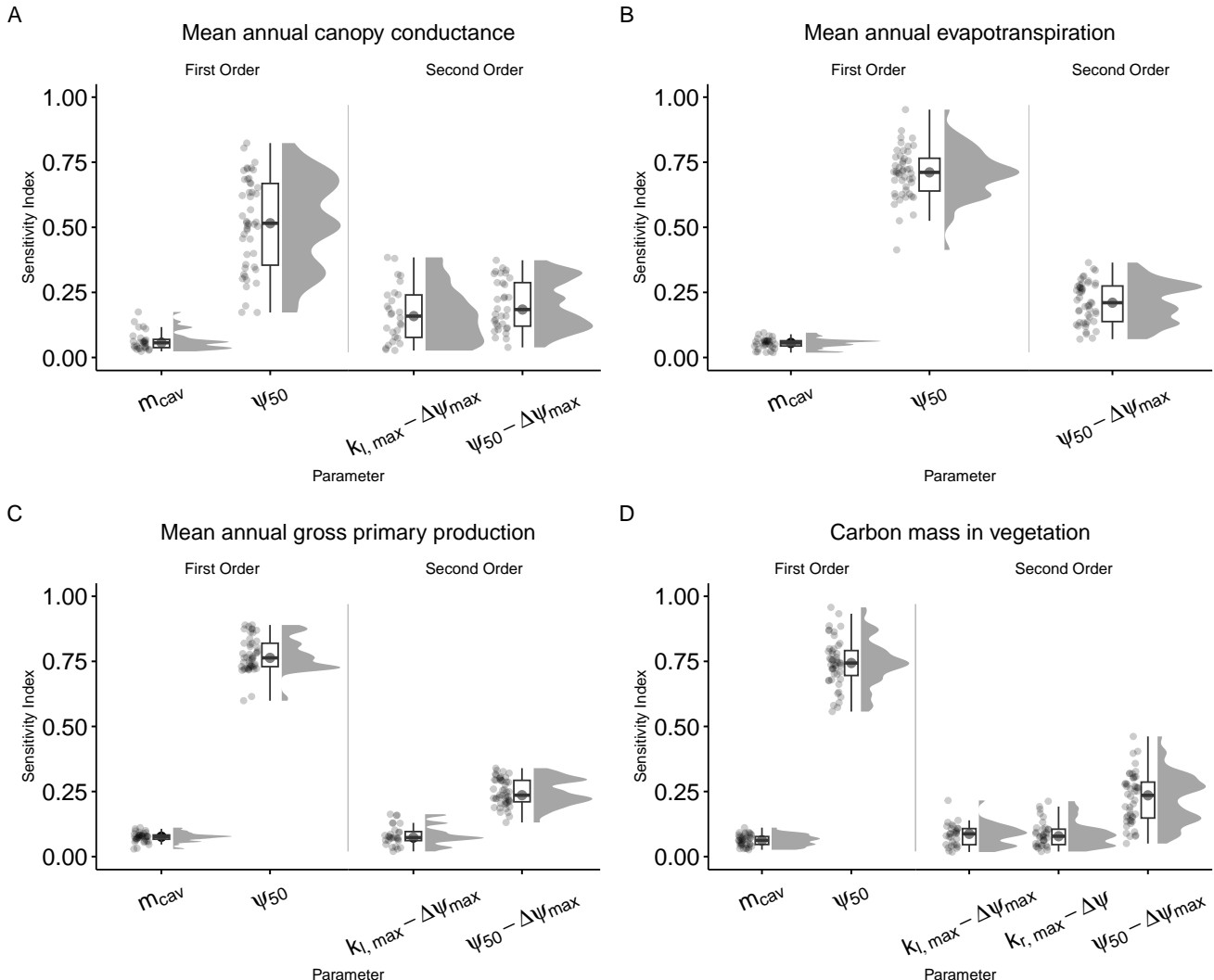

**Figure 3.** First and second order sensitivity indices for the seven parameters introduced in LPJ-GUESS-HYD. First-order indices indicate the sensitivity of model output solely due to variations of a single parameter. Second-order indices only consider variation in the output attributable to interactions between two parameters. First- and second-order indices are only shown for parameters with a median sensitivity greater than the median sensitivity of a dummy parameter (see methods for details). Each point represents the sensitivity index for a single species-site combination. The boxplots indicate the median and interquartile range of the sensitivity indices across species-site combinations. Each panel shows the sensitivity indices for a single model output (A) mean annual canopy conductance , (B) mean annual evapotranspiration, (C) mean annual gross primary productivity, (D) carbon mass in vegetation.

## 3.2  Evapotranspiration response to VPD

In LPJ-GUESS-HYD, evapotranspiration patterns of individual species were largely governed by the species-specific response to VPD (Figure 4, B). With increasing VPD classes, i.e. higher atmospheric demand for water, the spread of evapotranspiration patterns between species increased. While more isohydric species (e.g. *Pinus sylvestris*, *Abies alba*, *Populus tremuloides*) only marginally increased their evapotranspiration rate under higher VPD, more anisohydric species (e.g. *Fagus sylvatica*, *Quercus spec.*) tended to increase their evapotranspiration rates under higher VPD. In contrast, in LPJ-GUESS (Figure 4, C), although some species-specific differences in evapotranspiration rate were simulated, the general VPD response pattern was the same across all species; evapotranspiration increased with increasing VPD up to ~1.5 kPa and subsequently leveled off even as VPD continued to increase (Figure 4, right panel). Additionally, no clear pattern related to isohydricity was seen in LPJ-GUESS. Under high VPD the highest evapotranspiration rate was seen in an ostensibly more isohydric species, *Pinus sylvestris,* while the second highest rate was exhibited by *Quercus pubescens*, a relatively anishoydric species. Monospecific eddy-covariance flux sites were only available for a limited number of species (Figure 4, A). Here, more anisohydric species tended to continue transpiring even as VPD increased while more isohydric species reached maximum transpiration rates at relatively low levels of VPD and displayed decreasing evapotranspiration as VPD continued to increase. Under high VPD (~3 kPa) evapotranspiration simulated by LPJ-GUESS-HYD ranged from 0.9 mm d$^{-1}$ to 7 mm d$^{-1}$. The range in LPJ-GUESS was considerably smaller ranging from 1.4 mm d$^{-1}$ to 3.2 mm d$^{-1}$. For the eddy-covariance flux data observations at a VPD of ~3 kPa were only available for *Fagus sylvatica* which transpired 5.8 mm d$^{-1}$ at that VPD level.

## 3.3  Comparison of model results with observational data from eddy-covariance towers

The comparison of evapotranspiration simulated by LPJ-GUESS(-HYD) with evapotranspiration from the eddy covariance flux product in three pan-european drought years revealed contrasting results (Figure 5). Across all sites, species, and all drought years, the observed daily growing season evapotranspiration ranged from ~1.20 mm d$^{-1}$ to ~3.54 mm d$^{-1}$. LPJ-GUESS-HYD simulated a similar range (~1.14 - ~4.45 mm d$^{-1}$) while LPJ-GUESS simulated a narrower range (~1.50 - ~2.33 mm d$^{-1}$). Compared to the eddy covariance product both LPJ-GUESS and LPJ-GUESS-HYD displayed a similar level of mismatch with RMSEs of 0.70 and 0.84 mm d$^{-1}$, respectively. However, while LPJ-GUESS consistently underestimated the observed evapotranspiration (Mean signed deviation (MSD): -0.44), LPJ-GUESS-HYD showed a less negative bias (MSD: -0.06). For GPP both LPJ-GUESS and LPJ-GUESS-HYD show similar patterns broadly matching the observations. The RMSE for GPP was similarly low for both LPJ-GUESS and LPJ-GUESS-HYD at 0.0017 and 0.0021, respectively. For both model versions the MSD indicated no substantial over- or underestimation of the observations (LPJ-GUESS: -0.0003; LPJ-GUESS-HYD: -0.0007)

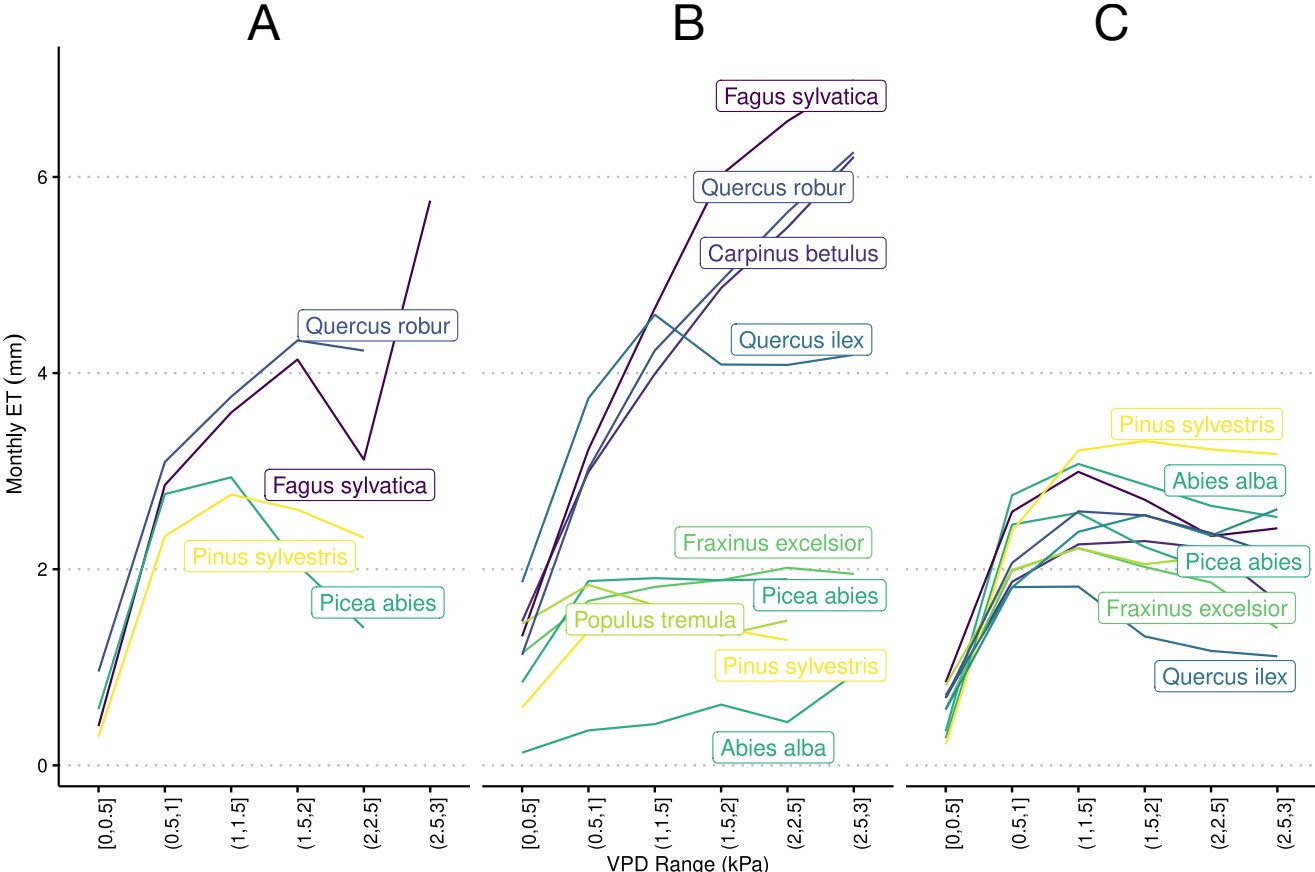

**Figure 4.** Species-specific daily evapotranspiration rates under differing levels of vapor pressure deficit from A) eddy-covariance flux towers, B) LPJ-GUESS-HYD, and C) standard LPJ-GUESS. The colors are ranked according to the $\lambda$ of each species (see Fig. 2) from high $\lambda$ (light) to low $\lambda$ (dark). Daily VPD was binned into 6 equally-sized classes representing increasing levels of drought. Species-specific responses to drought remain constant in LPJ-GUESS while clear differences between more anisohydric and more isohydric species are seen in LPJ-GUESS-HYD.

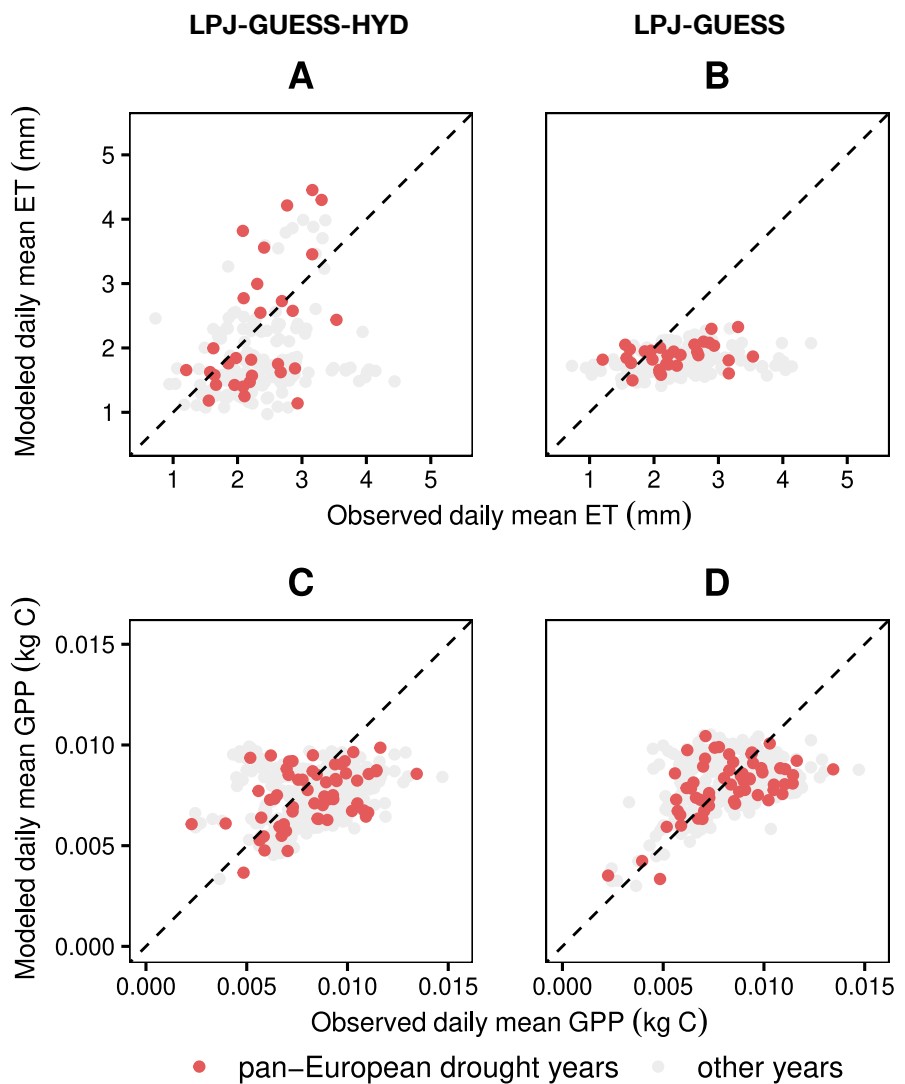

**Figure 5.** Comparison of measured eddy-covariance flux values of ET and GPP with modeled values from LPJ-GUESS-HYD (left column) and LPJ-GUESS (right column). LPJ-GUESS-HYD (A) matches observed ET patterns better than standard LPJ-GUESS (B) during three pan-European drought years while simulated GPP remains similar between both versions of the model (C,D). The dotted black line indicates perfect agreement between model and observations. Values above the dotted line represent instances where the model overestimates ET (GPP) compared to the observations and vice versa. Each dot corresponds to a single year and site and represents the average daily value over the growing season.

## 4 Discussion

We conducted an evaluation of the newly developed plant hydraulic architecture version of LPJ-GUESS, LPJ-GUESS-HYD, through a variance-based global sensitivity analysis and model evaluation for carbon and water fluxes at 34 eddy-covariance flux sites across Europe.

### 4.1 Relevance of hydraulic parameters

The results of our sensitivity analysis showed that of the seven newly introduced parameters (Table 1), two ($\psi_{50}$, $\Delta\psi_{max}$) consistently contributed substantially to the variance in model outputs either directly (Figure 3) or indirectly (Figure 2). Similarly, second-order interactions for all outputs included primarily those aforementioned parameters. Substantial differences in parameter importance were only seen for mean annual canopy conductance. Although, the two previously mentioned parameters still contributed the most to variance in simulated canopy conductance, nearly all other parameters played a substantial role as well (Figure 2, panel A). Additionally, across all sites and species, the sensitivity indices varied to a greater extent in the case of canopy conductance than in the other outputs (Figure 3, Figure 2). This pattern suggests that the relative influence of the new parameters is most evenly spread in model processes closely related (e.g., canopy conductance; see Figure 1) to the newly implemented plant hydraulic architecture. That is, while processes like carbon allocation to biomass which are further downstream from the new implementations are primarily affected by a single parameter ($\psi_{50}$), processes like canopy conductance which are directly affected by the new implementations are more sensitive to a greater number of the newly implemented parameters because the influence of these parameters is less diluted by other contributing model processes (e.g. plant demography) as is the case with the carbon allocation.

Strikingly, LPJ-GUESS-HYD output was by far most sensitive to variations in $\psi_{50}$, with roughly 75% of the variance in ET, GPP, and vegetation carbon being attributable to changes $\psi_{50}$ alone (Figure 3). While not directly comparable, this aligns both with a previous meta-analysis which suggested that $\psi_{50}$ was the single most effective predictor of tree drought mortality (Anderegg et al., 2016) and previous modeling efforts which indicated $\psi_{50}$ to substantially influence modeled xylem embolism Cochard et al. (2021). The meta-analysis additionally indicated that $k_{s,max}$ played little role in determining tree mortality due to drought (Anderegg et al., 2016), a result partially supported by our sensitivity analysis that showed $k_{s,max}$ has negligible influence on drought mortality. Although our analysis focused on water and carbon fluxes rather than outright mortality, these findings complement each other as they suggest that the traits that are responsible for impairing water transport and assimilation under drought stress are the same traits that ultimately determine whether a tree will experience drought damage or eventually die under prolonged drought.

The model's strong sensitivity to the maximum possible soil-to-leaf water-potential difference, $\Delta\psi_{max}$, is less intuitive. Along with the conductivity of roots, stem, and leaves, the soil-to-leaf water-potential difference, also referred to as the forcing pressure, plays a role in regulating the supply of water through the tree (Joshi et al., 2022; Da Sois

et al., 2024). Why, then, does the model sensitivity to $\Delta\psi_{max}$ overshadow the sensitivity to the parameters which govern conductivity, namely, $k_{r,max}$, $k_{s,max}$, and $k_{l,max}$? This divergent response can be explained by the relationship of $\Delta\psi_{max}$ and $\psi_{50}$ in LPJ-GUESS-HYD. Primarily, $\Delta\psi_{max}$ determines how tightly (or loosely) simulated leaf water potential is coupled to simulated soil water potential (Equation 12), affecting the degree of isohydricity. At a given soil water potential, species with a higher $\Delta\psi_{max}$ (i.e. looser coupling) will have a lower leaf water potential than species with a lower $\Delta\psi_{max}$ (i.e. stronger coupling). Due to the relationship between leaf water potential and xylem water potential in LPJ-GUESS-HYD (Equation 14), this means that the value of $\Delta\psi_{max}$, which influences leaf water potential (Equation 12), indirectly determines the xylem water potential and therefore affects the process of xylem cavitation. This is backed up by the significant second-order interactions between $\psi_{50}$ and $\Delta\psi_{max}$ (Figure 3). As Mcdowell et al. (2008) point out, the soil-to-leaf water potential difference, $\Delta\psi$, tends to increase with increasing transpiration until a critical xylem tension is reached leading to cavitation and, consequently, the hydraulic conductance approaches zero. It follows that as actual hydraulic conductance approaches zero, the maximum possible hydraulic conductance specified by $k_{r,max}$, $k_{s,max}$, and $k_{l,max}$ loses relevance. Indeed, this finding matches existing evidence from model sensitivity analyses indicating that parameters related to xylem safety and stomatal regulation explained a substantial fraction of the model variability, while whole plant conductance (i.e. $k_{r,max}$, $k_{s,max}$, and $k_{l,max}$ in our study) played a lesser role (Ruffault et al., 2022).

To reiterate, the results of the sensitivity analysis indicate that two of the hydraulic parameters introduced in LPJ-GUESS-HYD, namely $\Delta\psi_{max}$ and $\psi_{50}$ substantially shape long-term model behavior. These results imply that accurate estimations or, in the best case, measurements of these two parameters are paramount to reliably modeling plant hydraulics with LPJ-GUESS-HYD. Indeed, although using the, arguably better, parameterizations for $\psi_{50}$ from Martin-StPaul et al. (2017) (Figure A2) did not alter the general pattern that modeled evapotranspiration from LPJ-GUESS-HYD reflects the anisohydric-isohydric continuum, the evapotranspiration response to VPD of individual species was affected by this alternative parameterization (e.g. *Quercus ilex*; A2, panel B). What our sensitivity analysis cannot provide answers to, however, is how model sensitivity may change under stressed vs. non-stressed conditions. That is, does the pattern of influential parameters remain the same during drought as during non-drought conditions? To answer this, subsequent modeling endeavors specifically contrasting various climatic conditions are required.

## 4.2    Role of hydraulic architecture for carbon and water fluxes

The results of our sensitivity analysis show that simulated water and carbon fluxes from LPJ-GUESS-HYD are primarily influenced by hydraulic function – via $\psi_{50}$ – and secondarily by stomatal regulation – via $\Delta\psi_{max}$. These results are largely in line with findings from experiments and observations that repeatedly and consistently identify hydraulic failure as the preeminent factor governing tree drought mortality (Anderegg et al., 2016, 2015; Choat et al., 2012; Hammond et al., 2019; Adams et al., 2017).

However, the importance of $\Delta\psi_{max}$ in our model analysis also aligns with the ample evidence that stomatal regulation is critical in mediating drought responses of forests (Körner, 2019; Hajek et al., 2022; Mcdowell et al., 2008). The sensitivity of LPJ-GUESS-HYD to these widely supported mechanisms of tree drought response suggests that LPJ-GUESS-HYD should be able to correctly simulate drought and its associated impacts across a range of different species and hydraulic strategies.

To demonstrate the ability of LPJ-GUESS-HYD to model drought responses across hydraulic strategies, we analyzed the effect of increasing VPD on simulated evapotranspiration in both LPJ-GUESS-HYD and standard LPJ-GUESS (Figure 4). This analysis effectively showed that while LPJ-GUESS displayed nearly identical VPD response trajectories across all species, LPJ-GUESS-HYD exhibits distinct trajectories. This can be explained for one by the absence of VPD as a direct driver of evapotranspiration in standard LPJ-GUESS. Yet, it also shows the importance of the inclusion of dynamic stomatal regulation strategies as exhibited by the larger range in simulated evapotranspitation rates in LPJ-GUESS-HYD. More anisohydric species (i.e. lower $\lambda_{iso}$, higher $\Delta\psi_{max}$, see Table 2) tended to keep transpiring even under high VPD while more isohydric species displayed plateauing evapotranspiration as VPD increased. Our simulations revealed no distinct clustering of evapotranspiration responses to VPD, but rather a gradiation of responses dependent on the relevant parameters. This simulated behavior is congruent with the established notion of the anisohydric-isohydric continuum (Klein, 2014; Martínez-Vilalta et al., 2014; Martínez-Vilalta and Garcia-Forner, 2017). Similarly, the species-specific responses of evapotranspiration to VPD simulated by LPJ-GUESS-HYD reflect results from experiments identifying VPD as the most potent driver of both canopy conductance and evapotranspiration (Schönbeck et al., 2022; Flo et al., 2022). In particular, the order of the evapotranspiration-VPD response simulated by LPJ-GUESS-HYD (Figure 4) for *Fagus sylvatica*, *Quercus pubescens*, and *Quercus ilex* are comparable to the results from Schönbeck et al. (2022).

Lastly, to evaluate the efficacy of LPJ-GUESS-HYD at simulating the real world response of water and carbon fluxes to drought we compared simulated evapotranspiration and GPP with eddy-covariance fluxes from 34 sites across Europe during three pan-European drought years – 2003, 2015, and 2018 (Figure 5). Compared to LPJ-GUESS, LPJ-GUESS-HYD represents an improvement in terms of simulated evapotranspiration under drought. Since eddy-covariance flux data integrates the response of all species at a given site, our ability to conduct species-specific comparisons of modeled and observed evapotranspiration was limited. Nevertheless, the limited available data suggest that LPJ-GUESS-HYD is better at capturing the observed evapotranspiration patterns of more anisohydric species than of relatively isohydric species (Figure A1). This may also partially explain the underestimation of evapotranspiration by LPJ-GUESS-HYD seen at some sites (Figure 5, A), yet the limited availability of species-specific comparisons does not allow for a conclusive explanation. In any case, this indication together with the fact that $\Delta\psi_{max}$, which largely governs modeled stomatal regulation, was one of the most influential parameters, suggests that well-constrained estimates of $\Delta\psi_{max}$ are crucial for model performance.

Contrastingly, no meaningful difference was seen between LPJ-GUESS and LPJ-GUESS-HYD for simulated GPP under drought. Considering that the sensitivity analysis revealed that modeled GPP is sensitive to variations in $\psi_{50}$, the lack of differences between LPJ-GUESS-HYD and standard LPJ-GUESS may seem surprising. Yet, these results must be interpreted carefully. The control of $\psi_{50}$ on GPP in the sensitivity analysis stems from the fact that with high values of $\psi_{50}$ (i.e. low resistance to embolism) few viable parameter combinations remain, that is, $\psi_{50}$ represents a limiting factor which can override the effect of the other parameters. In the evaluation using the best estimate parameter sets (Table 2) the values of $\psi_{50}$ remain with a viable range. Additionally, despite lacking a mechanistic representation of photosynthetic response to drought the empirical relationships of photosynthesis to low water availability implemented in LPJ-GUESS – and, in fact, in a host of other DVMs – are rooted in reality and have been shown to be sufficient in reproducing past droughts and their effect on carbon uptake (Ciais et al., 2005; van der Woude et al., 2023; Gampe et al., 2021). Yet, the improved representation of evapotranspiration (based explicitly on canopy conductance) in LPJ-GUESS-HYD paves the way for implementing further hydraulic processes, such as capacitance, and improving existing ones, such as cavitation. Such advancements, coupled with sink-driven mechanisms (e.g. turgor-limited growth), are paramount to modeling carbon and water cycles in future climates where existing empirical relationships become less dependable (Körner, 2015; Torres-Ruiz et al., 2024).

### 4.3 Limitations of the modeling approach and ways forward

Despite the improvements in modeling plant-water relations offered by LPJ-GUESS-HYD, further improvements will be necessary in subsequent iterations of the model. Considering the hydraulic processes implemented in LPJ-GUESS-HYD (Figure 1), it is obvious that in the current state they are directed towards the water rather than the carbon cycle. As such, the path forward for LPJ-GUESS-HYD must focus on physiological processes connecting plant water-usage with plant carbon-usage, both in terms of carbon assimilation and carbon losses. One major source of carbon loss due to drought is tree mortality (Allen et al., 2010). In the current version of LPJ-GUESS-HYD, drought mortality is implemented based on xylem cavitation but not on the downstream ramifications of hydraulic failure (e.g.higher susceptibility to insects and other biotic agents) although these are generally considered to be significant secondary-drivers in drought induced mortality (Senf et al., 2020; Desprez-Loustau et al., 2006; Rouault et al., 2006; Bigler et al., 2006; Anderegg et al., 2015). Linking existing models dealing with biotic and non-biotic disturbance agents (Lagergren et al., 2012; Jönsson et al., 2012) to LPJ-GUESS-HYD could provide a pathway to better capture observed mortality associated with droughts. In this context, emphasis must be placed on mechanisms governing how drought stress increases vulnerability to these secondary processes. However, carbon losses due to drought are not confined only to tree mortality. Across the globe, an increase in drought-induced tree canopy dieback has been observed (Allen et al., 2010, 2015; Lloret et al., 2004; Frei et al., 2022; Carnicer et al., 2011; Hartmann et al., 2022). Evidence suggests that such dieback is caused primarily by hydraulic failure (Arend et al., 2022; Kannenberg et al., 2021; Walthert et al., 2021; Nolan et al., 2021), although a disruption of the soil-root interface (Körner, 2019; Carminati and Javaux, 2020) and preceding growth trends (Neycken et al., 2022) have been identified as

potential drivers as well. Regardless of the underlying cause, crown dieback reduces the leaf area, altering canopy water demand and growth even once the drought has subsided (Arend et al., 2022; Guada et al., 2016). While early

leaf senescence in response to drought has been widely observed in beech and other temperate broad-leaved species (Schuldt et al., 2020 and references therein), evidence suggests that coniferous species, such as spruce, may die from hydraulic failure before such protective measures can occur (Arend et al., 2021). Additionally, the relationship between drought intensity, hydraulic failure and early leaf senescence is difficult to quantify and studies establishing concrete thresholds for leaf senescence are scarce and focused on single species (e.g. Walthert et al., 2021). Nevertheless, early

leaf senescence plays an important role in governing tree drought response (Nadal-Sala et al., 2024). Yet, currently LPJ-GUESS(-HYD) neither includes any mechanistic nor empirical representation of this process. While the exact mechanisms may be too detailed for a model such as LPJ-GUESS-HYD, some relationship between hydraulic failure and reduced leaf area should be a part of future developments to ensure that the actual leaf area matches that which is able to be supported by the diminished sapwood area due to xylem cavitation.

Additionally, a better representation of drought-associated carbon losses (e.g. mortality, dieback, lost productivity, etc.) is only part of the puzzle. Most DVMs, including LPJ-GUESS-HYD, primarily model carbon allocation and tree growth as source-limited (Cabon and Anderegg, 2022; Eckes-Shephard et al., 2021). In LPJ-GUESS-HYD reduced carbon uptake under drought follows this pattern. As stomates close and gas exchange is reduced photosynthetic assimilation slows as well. Yet, emerging evidence emphasizes the importance of including sink limitations in models

as they a crucial factor in modulating tree growth, particularly during drought as cambial cell formation is limited by turgor (Körner, 2015; Peters et al., 2021; Cabon et al., 2020). While mechanistic turgor-driven growth models exist (Steppe et al., 2006; Génard et al., 2001; Peters et al., 2021) they are too complex, both temporally and physiologically, for direct implementation into LPJ-GUESS-HYD (Potkay et al., 2022). To bridge this gap, including plant water storage and hydraulic capacitance could be a starting point for a simple approximation of the more complex process

underlying turgor-driven growth limitations. Observations from dendrometers suggest that little to no growth occurs during periods of stem shrinkage, i.e., when plant water storage recedes (Zweifel et al., 2016). In contrast to dedicated turgor-driven growth models, the dynamics of plant water storage more easily lend themselves to implementation in DVMs and could nonetheless present a viable proxy for more complex sink-limitations under drought.

Lastly, if and when further developments to LPJ-GUESS-HYD are made subsequent sensitivity analyses should

be conducted. In this study, the sensitivity analysis focused on long-term model outputs such as annual water and carbon fluxes which are also commonly used for benchmarking DVMs(e.g. Seiler et al., 2022; Collier et al., 2018). However, as future developments discussed above (e.g. turgor-driven growth, drought-induced leaf-shedding) will likely focus on specific aspects of tree drought response, sensitivity analyses on finer temporal scales may be both more practical and useful than the larger yet coarser sensitivity analysis used here. To this end, future analyses could

also consider not only relying on direct model output variables but creating specific indices or metrics related to individual model processes (e.g. Ruffault et al., 2022).

## 5  Conclusion

In this study, we evaluated LPJ-GUESS-HYD for use with European tree species along an isohydricity gradient. The model was shown to simulate species-specific responses of evapotranspiration to increasing VPD in accordance with both results from experiments and current understanding of the anisohydric-isohydric continuum. A comparison of simulated ET and GPP with observations from eddy-covariance flux sites in three pan-European drought years (2003, 2015, 2018) revealed that LPJ-GUESS-HYD improved evapotranspiration compared to the standard version of LPJ-GUESS although both versions of the model displayed a similar fit of simulated to observed GPP. These results not only emphasize the importance of including mechanistic representations of plant hydraulic architecture in dynamic vegetation models but also highlight the fact that simulating both water- and carbon fluxes based on canopy conductance provides improvements in model performance compared to only using canopy conductance for the calculation of carbon fluxes. In this context, future developments of LPJ-GUESS-HYD should continue to focus on the connection between plant water-use and plant carbon-use, potentially under the aspect of sink-limited growth. Plant hydraulics are a crucial extension of current DVMs for modeling the effect of drought on altering ecosystem scale water-usage and continued refinements may be essential in providing robust estimates of future drought responses under a changing climate.

*Code and data availability.* LPJ-GUESS is publicly available at: https://doi.org/10.5281/zenodo.8065736. The version of LPJ-GUESS used in this study is publicly available at: https://doi.org/10.5281/zenodo.14000805. The model version presented here is identified by the commit hash 97c552c5. The analysis code used to produce the results and figures of this study is available at: https://doi.org/10.5281/zenodo.14001089

*Author contributions.* BM, AR, and CZ conceived the study. BM wrote the manuscript, conducted the model simulation runs, and conducted the data analysis. JD gathered and prepared input data for the model runs, contributed model code, and contributed to the draft writing. PP and KG contributed to model development. AB, QG, TG, and AK contributed to the interpretation of the results. DL and SA gathered an prepared parameter values for use in the sensitivity analysis. All authors edited the manuscript.

*Competing interests.* The authors declare no competing interests

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

## Appendix A: Appendix

*A1: Species-specific comparison of modeled evapotranspiration and evapotranspiration from eddy-covariance flux towers*

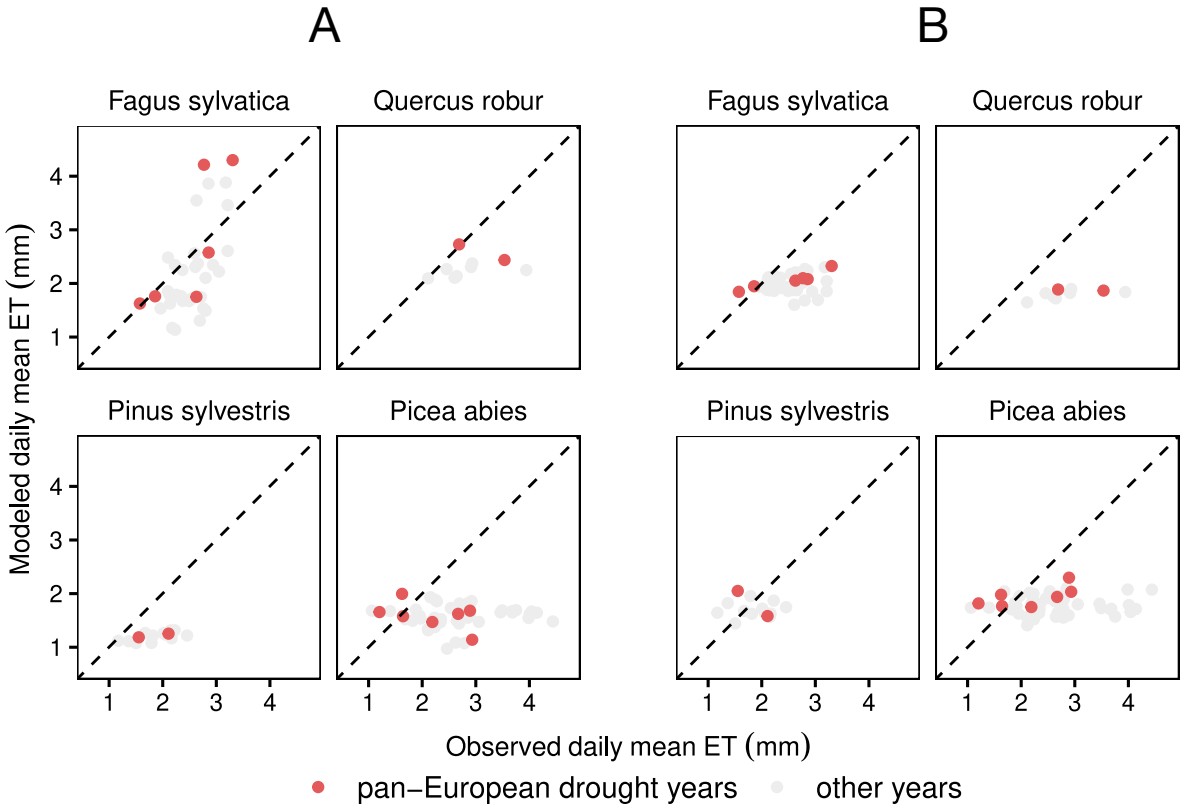

**Figure A1.** Of the 12 species analyzed in this study, monospecific eddy-covariance flux sites exist only for the four species shown here. In LPJ-GUESS-HYD (A) modeled ET better matches observed ET patterns for the relatively anisohydric species *Fagus sylvatica* and *Quercus robur* better than standard LPJ-GUESS (B) during three pan-European droughts. On the contrary, for the more isohydric species *Picea abies* and *Pinus sylvestris* both LPJ-GUESS-HYD (A) and LPJ-GUESS (B) underestimate observed ET. The dotted black line indicates perfect agreement between model and observations.

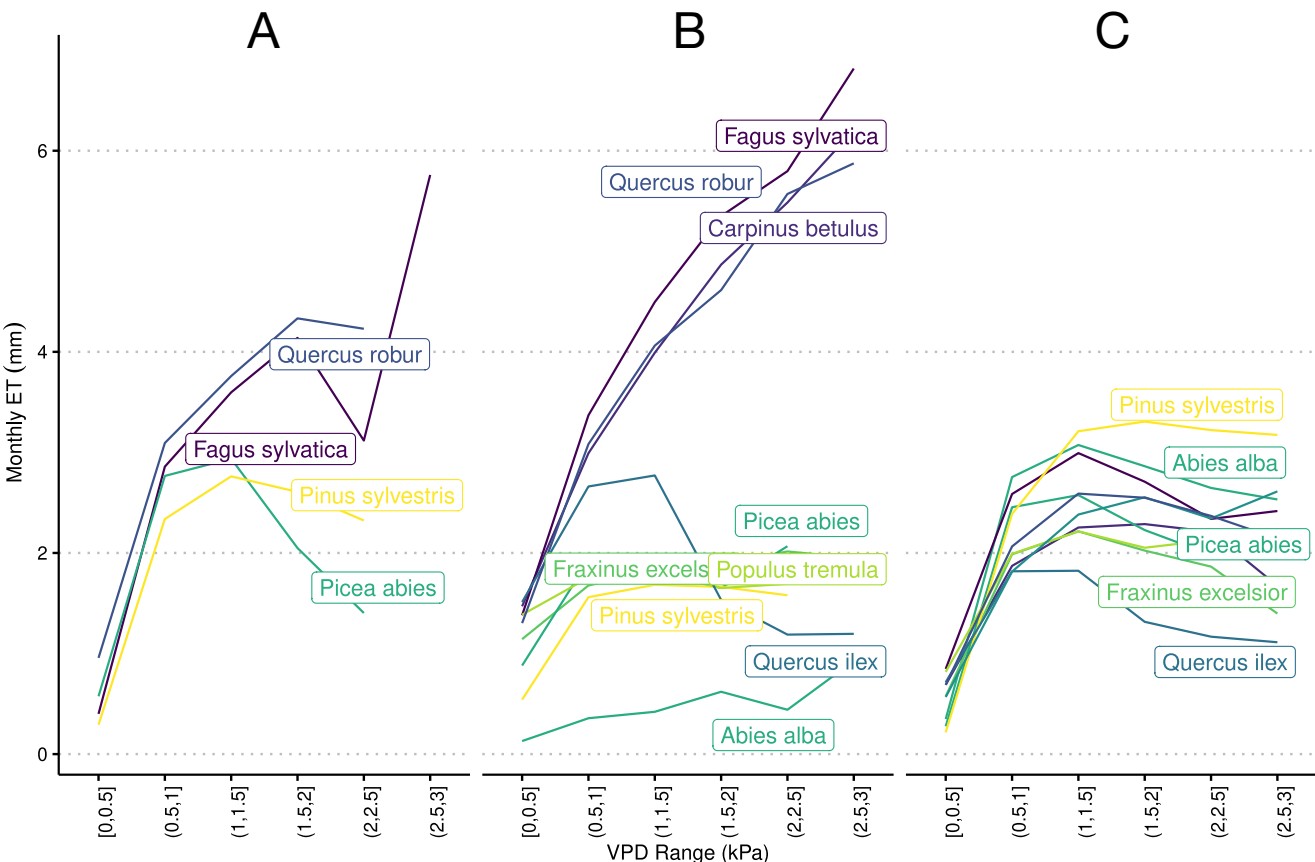

**Figure A2.** Species-specific daily evapotranspiration rates under differing levels of vapor pressure deficit from A) eddy-covariance flux towers, B) LPJ-GUESS-HYD, and C) standard LPJ-GUESS using $\psi_{50}$ values from Martin-StPaul et al. (2017). The colors are ranked according to the $\lambda$ of each species (see Fig. 2) from high $\lambda$ (light) to low $\lambda$ (dark). Daily VPD was binned into 6 equally-sized classes representing increasing levels of drought. Species-specific responses to drought remain constant in LPJ-GUESS while clear differences between more anisohydric and more isohydric species are seen in LPJ-GUESS-HYD.

*A3: Influence of soil water retention curve on $\psi_s$*

$$a = \frac{-e^{-4.396 - 0.0715 * \%\text{clay} - 4.88 * 10^{-4} * \%\text{sand}^2 - 4.285 * 10^{-5} * \%\text{sand}^2 * \%\text{clay}}}{10} \tag{A1}$$

and (see Saxton et al. (1986) Eq. 6):

$$b = -3.14 - 0.00222 * \%\text{clay}^2 - 3.484 * 10^{-5} * \%\text{sand}^2 * \%\text{clay} \tag{A2}$$