# Peer review of "Simulating the drought response of European tree species with the dynamic vegetation model LPJ-GUESS (v4.1, 97c552c5)"

_EGUsphere, 2024_

## Referee Report (RR1)

Review: "Simulating the drought response of European tree species with the dynamic vegetation model LPJ-GUESS (v4.1, 97c552c5)"

In this manuscript, the authors introduce, describe, and assess a new representation of plant drought response in the LPJ-GUESS DGVM. Specifically, the new work focuses on plant hydraulic architecture. The authors do not go so far as to mechanistically model xylem cavitation etc., but the empirical effects introduced here—designed in part to represent plant hydraulic *strategies*—are a good first step.

The manuscript has already received two peer reviews, and as the editor suggested, I focused mainly on the authors' responses to the previous reviewers. The authors have mostly addressed the requests from the first reviews and/or clarified why reviewers' suggestions were unnecessary in ways I agree with. I do have some follow-up points and new suggestions, however.

- L128: Authors' suggested revision in response to Reviewer 2's Comment 19 did not make it into the updated manuscript.
- L172 (Reviewer 2's Comment 26): "VDP" typo.
- L189 (Reviewer 2's Comment 27): Equations should be given, probably in the Appendix, for $A$ and $B$. This is much more appropriate in a *GMD* paper than referring interested readers somewhere else.
- L230-248 (Sect. 2.2.3): The authors did not add an equation here explaining "how the canopy conductance for plant hydraulic processes determines whether trees in LPJ-GUESS-HYD experience water limitation" as they said they would in their reply to Reviewer 2's Comment 2.
- Table 1 caption (L 274): "models" should be "model's."
- L 304-312 (Reviewer 2 comments 6, 8, and 34): I suggest adding a sentence here explicitly assuring readers that the Sobol' analysis accounts for potential collinearity.
- Fig. 5 (L 399):
  - "Monthly" isn't really a standard unit, since the number of days in a month can vary. It would be best (and consistent with Sect. 3.2) to use daily values instead (i.e., daily average over each month). It's fine for each point to still correspond to a calendar month.
  - Add model name to subplot titles.
- L 410-418 (Reviewer 2 Comment 33): I don't see why it should be unsurprising for "downstream" variables to be sensitive to fewer variables than "upstream" variables. I can understand why downstream variables would be *less* sensitive to each parameter, but that's not what you're saying here. And my understanding is that the Sensitivity Index between e.g. Fig. 4 A and B can't be compared, so my hypothesis can't be tested—is that right?

- L 458-467: Discussion of Fig. A2 should clarify that it's not just *any* alternate parameterization, but rather probably a *better* one (as discussed at L 325-330).
- L 542-548 (Reviewer 2 Comment 42): Please state explicitly that no mechanistic or empirical representation of this process is present in LPJ-GUESS or LPJ-GUESS-HYD.

-

---

## Author Response (AR2)

We thank the reviewer for providing valuable feedback and identifying some oversights we made in the previous round of reviews. Please find below our response to the reviewer's comments.

**1. L128: Authors' suggested revision in response to Reviewer 2's Comment 19 did not make it into the updated manuscript**

Done.

**2. L172 (Reviewer 2's Comment 26): "VDP" typo.**

Done.

**3. L189 (Reviewer 2's Comment 27): Equations should be given, probably in the Appendix, for $A$ and $B$. This is much more appropriate in a GMD paper than referring interested readers somewhere else.**

The equations are indeed in the Appendix but the in-text reference was misleading. Fixed.

**4. L230-248 (Sect. 2.2.3): The authors did not add an equation here explaining "how the canopy conductance for plant hydraulic processes determines whether trees in LPJ-GUESS-HYD experience water limitation" as they said they would in their reply to Reviewer 2's Comment 2.**

We thank the reviewer for identifying this oversight. We have added a brief explanation and the missing equation to the end of Section 2.2.3.

**5. Table 1 caption (L 274): "models" should be "model's."**

Done.

**6. L 304-312 (Reviewer 2 comments 6, 8, and 34): I suggest adding a sentence here explicitly assuring readers that the Sobol' analysis accounts for potential collinearity**

Done. We have added such a sentence on Line 318.

**7. Fig. 5 (L 399):**
**o "Monthly" isn't really a standard unit, since the number of days in a month can vary. It would be best (and consistent with Sect. 3.2) to use daily values instead (i.e., daily average over each month). It's fine for each point to still correspond to a calendar month.**
**o Add model name to subplot titles.**

We thank the reviewer for these suggestion and have adapted the plots accordingly.

**8. L 410-418 (Reviewer 2 Comment 33): I don't see why it should be unsurprising for "downstream" variables to be sensitive to fewer variables than "upstream" variables. I can understand why downstream variables would be less sensitive to each parameter, but**
**that's not what you're saying here. And my understanding is that the Sensitivity Index between e.g. Fig. 4 A and B can't be compared, so my hypothesis can't be tested—is that right?**

This is an interesting thought, however, as the reviewer states, this hypothesis unfortunately cannot be tested. The sensitivity indices represent the relative sensitivity of a given output variable to the input parameters and cannot be directly compared between different output variables. For example, we can conclude that, relative to the other parameters, $\lambda$ has a greater influence on canopy conductance than on carbon mass in vegetation, however, this does not necessarily mean that the absolute influence of $\lambda$ on canopy conductance is greater than it's absolute influence on carbon mass in vegetation.

Nevertheless, the reviewer's point stands that this behavior may not be as unsurprising as we initially stated. To address this we will remove the opening phrase "Perhaps unsurprisingly" from that sentence and emphasize that the sensitivity indices represent the relative influence for a given output.

**9. L 458-467: Discussion of Fig. A2 should clarify that it's not just any alternate parameterization, but rather probably a better one (as discussed at L 325-330).**

Done. We have change the sentence to "Indeed, although using the, *arguably better*, parameterizations for $\psi_{50}$ from *Martin-StPaul et al., (2017)* (Figure A2) did not alter ..."

**10. L 542-548 (Reviewer 2 Comment 42): Please state explicitly that no mechanistic or empirical representation of this process is present in LPJ-GUESS or LPJ-GUESS-HYD.**

Done. We have added the following sentence starting on Line 544:

"Yet, currently LPJ-GUESS(-HYD) neither includes any mechanistic nor empirical representation of this process."

---

## Author Response (AR3)

**Just one more thing, on Fig. 5: Do the plotted points still represent the average daily value in each month? That's fine, of course, but please explain that in the caption.**

We thank the editor for taking the time to checking our revisions and noting the ambiguity in Figure 5. We have clarified in the caption that each point corresponds to a single year and site and represents the average daily value for the growing season.